# Micronutrient status and dietary patterns among children with autism in Central Vietnam: A cross-sectional baseline survey to inform targeted intervention

Huyen Thu Doan[1,2]*, Ngoc Thi Dieu Phan[1], Hoa Thi Ho[2], Lieu Thi Thu Nguyen[1], Khanh Nam Do[1], Oanh Thi Phuong Nguyen[1], Ngoc Bao Trinh[2]

**1** Hanoi Medical University, Hanoi, Vietnam, **2** Institute of Nutrition Research and Development, Hanoi, Vietnam

* doanhuyen3001@gmail.com

## Abstract

Children with autism spectrum disorder (ASD) in low- and middle-income countries may face multiple forms of malnutrition, but data to guide targeted nutrition interventions are limited. We conducted a cross-sectional baseline survey to inform a micronutrient-focused intervention in 48 children aged 2–9 years diagnosed with ASD, recruited from five intervention centers in Central Vietnam. Caregivers completed a 24-hour dietary recall and a one-month food frequency questionnaire. Anthropometric measurements were collected and converted to World Health Organization z-scores. Fasting blood samples were analyzed for hemoglobin, serum ferritin, and serum zinc concentrations. Non-parametric tests and bootstrap confidence intervals were used to compare preschool-aged children (<5 years) with those 5–9 years. Undernutrition was identified in 16.7% (underweight) and 20.8% (stunting), while 6.3% of participants were overweight or obese. Zinc deficiency affected 45.8% of children, low ferritin was found in 16.7%, and anemia in 10.4%. Two concurrent micronutrient deficiencies were present in 16.7%. The median energy intake met 84.6% of national recommendations. Dietary fiber intake was universally inadequate, and most children consumed less than the recommended levels of iodine (97.0%), zinc (64.6%), vitamin C (64.6%), and calcium (56.3%). Older children were significantly more likely to have inadequate calcium intake than younger ones (73.9% versus 40.0%). Dietary patterns were dominated by cereal-based foods, with infrequent intake of legumes, vegetables, dairy, and animal-source foods. These findings reveal a triple burden of undernutrition, micronutrient deficiencies, and emerging overnutrition among children with ASD. The results underscore the urgent need for early nutritional screening and dietary improvement strategies. These baseline data offer critical evidence for designing context-appropriate nutrition interventions in similar low-resource settings.

**Data availability statement:** The minimal dataset underlying the findings of this study is not publicly available due to ethical restrictions and participant privacy concerns involving children with autism spectrum disorder. However, de-identified data are available upon reasonable request for the purpose of research replication. Requests for data access should be directed to the Data Access Committee of the Institute of Nutrition Research and Development (INRD), an independent non-author institutional body, via its official institutional email (nghiencu-uvaphattriendinhduong@gmail.com). This institutional contact is responsible for reviewing data access requests and ensuring long-term data storage and availability. All study data are securely stored on password-protected institutional servers at INRD and are managed in accordance with institutional data governance policies and applicable national regulations.

**Funding:** This work was supported by the Master, PhD Scholarship Programme of Vingroup Innovation Foundation (VINIF) (Grant No. VINIF.2024.ThS.74 to HTD). The funders had no role in study design, data collection and analysis, decision to publish, or preparation of the manuscript.

**Competing interests:** The authors have declared that no competing interests exist.

## Introduction

Autism spectrum disorder (ASD) is a lifelong neurodevelopmental disorder characterized by persistent difficulties in social communication and interaction, together with limited and repetitive behavior. Global estimates place prevalence close to 1% of children, and the most recent United States surveillance reports 1 in 36 eight-year-olds, underlining the growing public health impact of ASD [1,2]. The prevalence of ASD in Vietnamese children aged 18–30 months was estimated to be approximately 0.76 percent [3].

Studies have shown that children with ASD face a dual burden of malnutrition, with studies reporting both undernutrition and overweight compared to their neurotypical peers [3–5]. Children with ASD often have feeding problems associated with sensory-driven selective eating, gastrointestinal disorders and reduced feeding capacity, which may limit the variety of foods and compromise the absorption of nutrients [6–8]. In low- and middle-income countries (LMICs) such as Vietnam these problems are exacerbated by the lack of specialized nutrition services, limited availability of fortified foods and the lack of biochemical screening for micronutrient deficiencies [9].

Although the nutritional status of children with ASD has been highlighted in Vietnam, the existing evidence is still fragmentary. Previous research, such as the study in Nghe An Province, focused mainly on anthropometric results [10]. However, integrated data linking physical growth to dietary intake patterns and micronutrient biomarkers are lacking. This methodological gap prevents a comprehensive understanding of the specific dietary risks of regions where socio-economic conditions are different from those in metropolitan areas. National 2024 statistics show that the North Central Coast, including Nghe An, is facing major socio-economic challenges, with per capita income almost half that of economic centers such as the South [11]. These differences, combined with the uneven distribution of doctors, limit regional access to specialized nutritional support and highlight the urgent need for targeted research.

To address these evidence gaps in evidence, this study provides a comprehensive baseline assessment of the nutritional status of ASD children in central Vietnam. By examining the interaction between diet, physical growth and micronutrient biomarkers, we are aiming to create a holistic nutrition profile for this vulnerable population. In particular, the study aimed to:

1. Assess the anthropometric nutritional status of children with ASD using WHO standards to determine the prevalence of stunting, underweight, and overweight/obesity.

2. Quantify micronutrient biomarkers (hemoglobin, ferritin, and serum zinc) to evaluate the prevalence of biochemical deficiencies.

3. Describe dietary intake and consumption patterns through 24-hour recalls and food frequency data.

4. Compare these indicators between preschool-aged (<5 years) and school-aged children (5–9 years) to identify age-specific nutritional risks during the transition to formal schooling.

## Materials and methods

### Design, population, and settings

**Conceptual framework.** This study employs a mechanistic framework to investigate the cascading impact of ASD-specific challenges on nutritional health (Fig 1). We conceptualize that ASD-related feeding challenges, specifically sensory-driven selective eating and gastrointestinal distress, act as primary drivers of disrupted dietary patterns. These behaviors create a restrictive nutritional environment characterized by low dietary diversity and chronic inadequacy of essential micronutrients (zinc, fiber, and calcium). We hypothesize that these dietary gaps are the physiological precursors to biochemical deficiencies (low serum zinc and ferritin), which ultimately manifest as anthropometric deficits (stunting and underweight). By integrating sociodemographic constraints (income and caregiver limitations) as background moderators, this framework shifts the focus from isolated observations to a mechanistic pathway, explaining how behavioral phenotypes in children with ASD in Central Vietnam translate into long-term physical growth impairments across the preschool-to-school-age transition.

**Study design and setting.** This cross-sectional study was conducted from June to September 2024 in five autism intervention centers located in Nghe An province, Vietnam. The study area comprises both urban and predominantly rural districts. The study involved five autism intervention centers that provide rehabilitation and educational services for children with neurodevelopmental disorders.

**Selection of intervention centers.** The five centers were selected using a convenience-based participatory approach. All identifiable and accessible intervention centers in Nghe An province were formally invited, of which five centers provided institutional approval to participate. These facilities primarily deliver intervention services for children with ASD and related conditions (e.g., ADHD, learning difficulties). These centers provide intervention but not diagnostic services.

**Autism diagnosis and referral pathway.** All participating children had confirmed clinical diagnosis of ASD made at specialized pediatric or psychiatric hospitals prior to enrolment. Diagnostic procedures followed the Diagnostic and Statistical Manual of Mental Disorders, Fifth Edition (DSM-5) criteria and were independent of the intervention centers [12]. Children were referred to these centers solely for rehabilitation and support services following their clinical diagnosis.

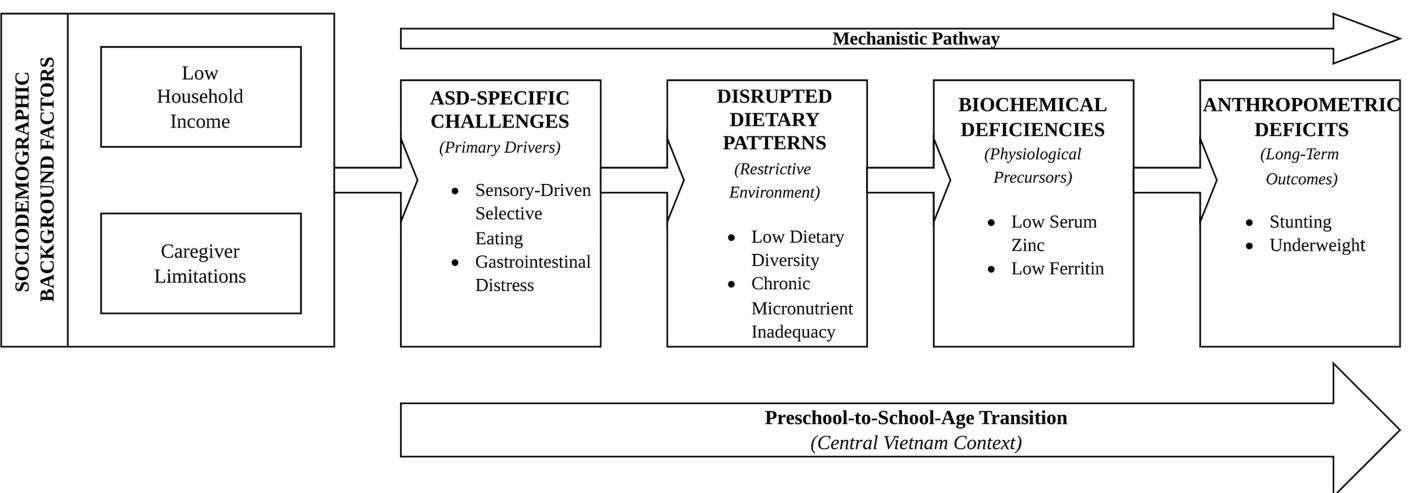

**Fig 1. Mechanistic framework of the nutritional and growth outcomes in children with ASD.** The model illustrates the cascading impact of ASD-specific behavioral and physiological drivers (sensory-driven selective eating and gastrointestinal distress) on dietary patterns and biochemical status. Socioeconomic factors (household income) and caregiver constraints act as moderators within this pathway, which ultimately culminates in anthropometric deficits such as stunting and underweight. ASD: Autism Spectrum Disorder.

**Study participants.** Participants were (1) children aged 2–9 years who were enrolled at one of the five centers, had a clinician-confirmed Diagnostic and Statistical Manual of Mental Disorders, Fifth Edition (DSM-5) diagnosis of autism spectrum disorder, and (2) their primary caregiver (e.g., parents or grandparents) responsible for the child's daily feeding [12].

Inclusion criteria comprised: (1) children present at the centers during the survey period; (2) a primary caregiver available and willing to participate; and (3) approval from the center's leadership. Exclusion criteria included: (1) children experiencing acute illness during the study period; (2) absence from the intervention center; and (3) caregivers with psychiatric disorders, cognitive impairment, or hearing/language difficulties preventing effective communication.

## Sample size

This study employed a census-based approach, inviting all 48 eligible children aged 2–9 years from the five participating centers. As an exploratory study, this sample size aimed to provide preliminary prevalence estimates and detect nutritional differences between age groups. Detailed calculations regarding statistical power and precision are provided in S1 File.

## Data collection procedures

**Sociodemographic and perinatal data collection.** Basic information on the child and their primary caregiver was collected using a structured questionnaire administered during in-person interviews. Collected variables included the child's age, sex, birth order, number of children in the family, age at autism diagnosis (in months), and duration of intervention received. Perinatal characteristics such as gestational age at birth and birth weight were also recorded [13]. Gastrointestinal symptoms were assessed based on caregiver report, including frequency of constipation (defined as ≥2 episodes/week) and diarrhea (defined as ≥3 episodes/day) in the past month [14]. While these definitions were not based on standardized diagnostic tools, they reflect practical criteria used in similar caregiver-reported studies in resource-limited settings. Socioeconomic and contextual factors included caregiver's educational attainment (university level or higher), monthly household income (categorized by the national poverty threshold of 8 million VND), and study site location (urban vs. rural). All data were recorded in a standardized form and double-entered for quality control.

**Anthropometric measurements.** Weight was measured to the nearest 0.1 kg using the Tanita HD-351 digital pediatric scale (Tanita Corp., Tokyo, Japan), and height to the nearest 0.1 cm using a ShorrBoard portable stadiometer (Shorr Productions, Olney, MD, USA), following standard World Health Organization (WHO). Measurements were taken in the morning, with children in light clothing.

Z-scores for weight-for-age (WAZ), height-for-age (HAZ), weight-for-height for children under 5 years (WHZ), and BMI-for-age (BAZ) for those aged 5–9 years were calculated using WHO Anthro v3.2.2 and AnthroPlus v1.0.4 software (WHO, Geneva, Switzerland) according to 2006 WHO growth standards [15]. Classification was based on standard cut-offs:

- Underweight: WAZ < −2 SD
- Stunting: HAZ < −2 SD
- Wasting: WHZ (children <5 years); BAZ (children 5–9 years) < −2 SD
- Overweight: WHZ > +2 SD; BAZ > +1 SD
- Obesity: WHZ > +3 SD; BAZ > +2 SD

**Gastrointestinal symptoms (operational definitions).** Gastrointestinal symptoms were assessed based on caregiver report using predefined operational criteria. Constipation was defined as fewer than three bowel movements per week and/or the passage of hard or difficult-to-pass stools during the past month. Diarrhea is defined as the passage of 3 or more loose or liquid stools per day (or more frequent passage than is normal for the individual) [16].

**Dietary assessment. 24-Hour dietary recall.** Two registered dietitians, trained and standardized in the USDA Automated Multiple-Pass Method (five passes; inter-observer technical error < 5%), interviewed each child's primary caregiver, and the class teacher when center-provided meals were involved, to record every food and beverage consumed between 00:00 and 24:00 [17].

A Vietnamese recall form that subdivides the day into six eating occasions (breakfast, morning snack, lunch, afternoon snack, dinner, and a bedtime snack) was used; the blank form and interviewer guide are supplied as S2 File.

Portion sizes were estimated with the National Institute of Nutrition (NIN) photographic food-portion atlas plus a set of local household utensils (rice bowl, soup spoon, 200 mL cup) calibrated with water to the nearest 5 mL. Mixed dishes were matched with nutrient values from standardized Vietnamese food composition data. If a dish was absent, its raw ingredients and edible fractions were taken from the same table, and nutrient values were recalculated with the 2007 Vietnamese Food Composition Table [18].

All dietary records were processed with Vietnam Eiyokun software (version 5.0; NIN, Hanoi, Vietnam). Intakes were then benchmarked against the age- and sex-specific Recommended Nutrient Intakes 2016 issued by the Ministry of Health [19]. Inadequate intake was operationally defined as <75% of the age- and sex-specific recommended dietary allowance (RDA). This pragmatic cut-off was adopted because (1) the study relied on single 24-hour dietary recalls, precluding the use of usual intake distributions or EAR-based probability methods; and (2) the < 75% RDA threshold has been widely applied in pediatric and small-sample nutrition studies as a conservative screening criterion to identify children at potential risk of inadequate intake rather than to diagnose true deficiency.

**Food Frequency Questionnaire (FFQ).** Habitual intake of eight core food groups - cereals; meat/poultry; fish; eggs; dairy; legumes/nuts; fruit; and vegetables - during the previous month was assessed with a semi-quantitative FFQ adapted from the validated NIN pediatric FFQ [20]. Caregivers selected one of six frequency categories (Daily; 4–6 times/ week; 1–3 times/week; 1–3 times/month; < 1 time/month; Never). The full questionnaire, including all food groups and frequency categories, is available in S3 File.

**Data-quality control**. Ten percent of recalls were independently repeated on the same day; the mean absolute difference in reported energy was 6%. Principles of Nutritional Assessment All dietary records were screened for implausible totals (< 2 × basal metabolic rate) and duplicate items before analysis.

**Biochemical indicators.** Venous blood (3 mL) was drawn from each child between 08:00 and 10:00 after an overnight fast of at least 8 h (caregivers confirmed the child had not consumed any food or beverages other than plain water since 22:00 the previous night). All blood samples were collected by certified laboratory technicians at Nghe An Obstetrics and Pediatrics Hospital, following standard procedures for trace-element collection in young children. Samples were collected into trace-element–free vacutainers (BD Royal Blue, no additive) and transported on ice to Laboratory Department of Nghe An Obstetrics and Paedtrics Hospital within 2 hours.

Serum was separated at 3 000 g for 10 min and analyzed the same morning; aliquots that could not be processed immediately were stored at 193.15 K (< 2 weeks).

Quality control procedures were carried out during the pre-analytical, analytical, and post-analytical phases in accordance with the national quality requirements of the Ministry of Health (Decision No. 2429/QĐ-BYT) [21]. Daily internal quality control (IQC) and routine national external quality assessment (EQA) programs were implemented to ensure analytical precision and inter-laboratory comparability. For the analysis of serum zinc, instruments were calibrated daily, commercial control materials were analyzed with each batch, and all tests were performed by trained personnel according to standardized operating procedures.

Hemoglobin (Hb): Measured using cyanmethemoglobin method (Beckman Coulter DxH 600); anemia defined as Hb < 110 g/L (<5 years) or <115 g/L (5–9 years) [22].

Serum Ferritin: Measured by immunoturbidimetric assay (Beckman AU680); iron deficiency defined as <12 μg/L (<5 years) or <15 μg/L (5–9 years) [23].

Serum Zinc: Measured by flame atomic absorption spectrometry (Shimadzu AA-7000). Low serum zinc concentration was defined as serum zinc <65 µg/dL for children under 10 years of age, in accordance with World Health Organization (WHO) and International Zinc Nutrition Consultative Group (IZiNCG) recommendations [24].

## Statistical analysis

Children were stratified into two age groups (<5 years vs. 5–9 years) based on both developmental and programmatic considerations. In Vietnam, national guidelines distinguish the "golden period" of early childhood (under 5 years), characterized by rapid neurodevelopment and high nutritional sensitivity, from school-age children (5–9 years). This division also reflects the structure of intervention centers, where goals shift from early feeding skill acquisition to school readiness and adaptive functioning. Accordingly, these comparisons were conducted to explore nutritional variations across these distinct developmental stages.

Data were analyzed using IBM SPSS Statistics version 27.0 (IBM Corp., Armonk, NY, USA). Statistical significance was set at $p < 0.05$. Given the exploratory nature of the study and the limited sample size ($n = 48$), analyses were primarily descriptive and comparative. While multivariable adjustments for sex, socioeconomic status, and center location were considered, they were not performed to avoid model overfitting and unstable estimates. These factors are instead reported descriptively to provide context for the findings.

Continuous outcomes were compared with two-tailed Mann–Whitney U tests (effect size r), and binary outcomes with Fisher's exact test; absolute risk differences (RD) and 95% confidence intervals (CI) were reported. Because of the small sample ($n = 48$), all CIs for medians, proportions and RD were obtained with bias-corrected and accelerated (BCa) bootstrap resampling (2 000 iterations, boot package). Effect size ($r$) interpretations follow standard thresholds (0.1 = small, 0.3 = medium, 0.5 = large) [25].

## Ethics statement

The study was approved by the Institutional Review Board of Hanoi Medical University (Approval No. 1487/GCN-HMUIRB). Written informed consent was obtained from the caregivers of all participating children with autism spectrum disorder prior to data collection. All data were collected and analyzed in de-identified form to ensure participant confidentiality.

## Results

A total of 48 children with ASD were included, comprising 25 under 5 years and 23 aged 5–9 years. The majority were boys (75.0%), and most families had two children (median: 2 [IQR: 1.25–3.0]). Median age at ASD diagnosis was 25.5 months, and median intervention duration was 23 months. Notably, 72.9% of households reported a monthly income below 8 million VND (Table 1).

Table 2 presents the distribution of anthropometric z-scores among Vietnamese children with ASD, stratified by age group. Median WAZ was slightly lower in school-age children (-1.26) than in preschoolers (-0.92), but the difference was not statistically significant ($p = 0.984$, $r = 0.003$). Median HAZ was -1.24 in the younger group and –0.84 in the older group ($p = 0.634$). WHZ and BAZ were not compared between groups due to age-specific applicability.

Table 3 presents the prevalence of undernutrition and overnutrition among children with ASD by age group. Overall, 16.7 percent of children were underweighted, 20.8 percent stunted and 12.6 percent met the criteria for stunting and overweight or obesity. Compared to the national data for neurotypical peers, the study population had a higher prevalence of undernutrition (stunting: 20.8% vs. 19.6%; underweight: 16.7% vs. 11.5–12.2%) but a lower prevalence of overweight/obesity (12.6% vs. 19.0%) [26]. No statistically significant differences in prevalence were observed between preschool-aged and school-aged children ($p > 0.05$).

**Table 1. Sociodemographic and perinatal characteristics of children with autism spectrum disorder and their caregivers (_n_ = 48).**

| Variable | < 5 years (_n_ = 25) | 5 − 9 years (_n_ = 23) | Total (_n_ = 48) |
|---|---|---|---|
| Age (years) | 3.0 (3.0–4.0) | 5.0 (5.0–6.5) | 4.0 (3.0–5.0) |
| Boys | 20/25 (80.0) | 16/23 (69.6) | 36/48 (75.0) |
| Birth order | | | |
| 1st | 11/25 (44.0) | 9/23 (39.1) | 20/48 (41.7) |
| 2nd | 9/25 (36.0) | 8/23 (34.8) | 17/48 (35.4) |
| ≥3rd | 5/25 (20.0) | 6/23 (26.1) | 11/48 (22.9) |
| Number of children in family | 2.0 (1.0-2.0) | 2.0 (2.0-3.0) | 2.0 (1.25–3.0) |
| Age at diagnosis (months) | 24.0 (21.0-30.0) | 28.0 (24.0–36.0) | 25.5 (23.25-30.0) |
| Duration of intervention (months) | 16.0 (11.0–25.5) | 25.0 (12.0-52.0) | 23.0 (12.0– 30.0) |
| Gestational age < 37 weeks | 2/25 (8.0) | 4/23 (17.4) | 6/48 (12.5) |
| Birth weight < 2500g | 2/25 (8.0) | 3/23 (13.0) | 5/48 (10.4) |
| Constipation (≥2 times/week) | 20/25 (80.0) | 20/23 (87.0) | 40/48 (83.3) |
| Diarrhea (≥3 times/day) | 12/25 (48.0) | 16/23 (69.6) | 28/48 (58.3) |
| Household income <8 million VND | 15/25 (60.0) | 20/23 (87.0) | 35/48 (72.9) |
| Caregiver's education ≥ University | 9/25 (36.0) | 11/23 (47.8) | 20/48 (41.7) |
| Study site | | | |
| Urban | 19/25 (76.0) | 18/23 (78.3) | 37/48 (77.1) |
| Rural | 6/25 (24.0) | 5/23 (21.7) | 11/48 (22.9) |

VND: currency of Vietnam. Data are reported as median (inter-quartile range) for continuous variables and n (%) for categorical variables.

**Table 2. Anthropometric z-scores in children with ASD, by age group (_n_ = 48).**

| Indicators | < 5 years (_n_ = 25) Median (IQR) | 5 − 9 years (_n_ = 23) Median (IQR) | BCa 95% CI (Bootstrap) | p-value | Effect size (r) | Mann-Whitney U |
|---|---|---|---|---|---|---|
| WAZ | -0.92 (-1.24; -0.74) | -1.26 (-1.90; 0.18) | -1.26; -0.74 | 0.984 | 0.003 | 286.5 |
| HAZ | -1.24 (-1.69; -0.47) | -0.84 (-1.86; -0.15) | -1.50; -0.64 | 0.634 | 0.07 | 264.0 |
| WHZ | -0.29 (-0.92; 0.38) | NA | -0.33; 0.18 | NA | NA | NA |
| BAZ | NA | -0.65 (-0.93; 0.87) | -0.85; -0.11 | NA | NA | NA |

NA: Not applicable; WAZ: Weight-for-age Z-score WAZ; HAZ: Height-for-age Z-score; WHZ: Weight-for-height; BAZ: BMI-for-age. Continuous data are presented as median (IQR) and compared using Mann–Whitney U tests.

**Table 3. Prevalence of undernutrition and overnutrition among children with ASD, by age group (_n_ = 48).**

| Indicators | < 5 years (_n_ = 25) | 5 − 9 years (_n_ = 23) | Total (_n_ = 48) | p-value | Risk difference RD (95% CI) |
|---|---|---|---|---|---|
| Underweight (WAZ < -2SD) | 4/25 (16.0) | 4/23 (17.4) | 8/48 (16.7) | >0.99 | -1.4 (-22.5; 20.0) |
| Stunting (HAZ < -2SD) | 5/25 (20.0) | 5/23 (21.7) | 10/48 (20.8) | >0.99 | -1.7 (-24.7; 21.3) |
| Wasting (WHZ or BAZ < -2SD) | 1/25 (4.0) | 2/23 (8.7) | 3/48 (6.3) | 0.545 | -4.7 (-18.5; 9.1) |
| Overweight/Obesity (WHZ or BAZ > 2SD) | 1/25 (4.0) | 2/23 (8.7) | 3/48 (6.3) | 0.545 | -4.7 (-18.5; 9.1) |

WAZ: Weight-for-age Z-score WAZ; HAZ: Height-for-age Z-score; WHZ: Weight-for-height; BAZ: BMI-for-age. Differences in prevalence between age groups were assessed using Fisher's exact test. Risk differences (RD) and 95% confidence intervals were calculated using the Newcombe method (score-Wilson).

Table 4 presents median concentrations of hemoglobin, ferritin, and zinc in children with ASD, stratified by age group. Hemoglobin levels were significantly higher in school-aged children (126.0 vs. 120.0 g/L; $p = 0.024$), with a small-to-moderate effect size ($r = 0.33$). Although ferritin levels were also higher in older children, the between-group difference did not reach statistical significance ($p = 0.055$). Zinc concentrations were similar across groups, showing no significant differences by age.

Table 5 summarizes the prevalence of anemia, iron deficiency, low serum zinc, and combined deficiencies among children with ASD across two age groups. Overall, 10.4% had anemia, 16.7% had low ferritin, and 45.8% had low serum zinc. No statistically significant differences were found between age groups in any indicator ($p > 0.05$). The results suggest a substantial burden of micronutrient deficiency in this population, with a notable proportion presenting multiple deficiencies.

Median energy intake was 4 676 kJ/day, meeting 84.6% of RDA on average. The most frequently inadequate nutrients were fiber (100.0%), iodine (97.0%), zinc (64.6%), vitamin C (64.6%), and calcium (56.3%). Protein intake was largely adequate, with only 6.3% of children below the RDA. Over 40% had sodium intake exceeding recommendations (Table 6).

Children aged 5–9 years had a significantly higher prevalence of inadequate calcium intake than those <5 years (73.9% vs. 40.0%; RD: -33.9 [-60.1; -7.6], $p = 0.037$). Similar trends were observed for vitamin B2 and magnesium, although not statistically significant. Complete data is provided in S1 Table.

Food frequency analysis showed all children consumed cereals daily. However, 45.9% rarely or never consumed legumes and nuts, 58.3% consumed few other vegetables, and 22.9% consumed dairy infrequently. These patterns may explain the high prevalence of micronutrient inadequacy. Details are presented in S2 Table.

**Table 4. Biochemical indicators of nutritional status in children with ASD, by age group ($n = 48$).**

| Indicator | < 5 years ($n = 25$) | 5 – 9 years ($n = 23$) | BCa 95% CI (Bootstrap) | p-value | Effect size (r) | Mann-Whitney U |
|---|---|---|---|---|---|---|
| Hemoglobin (g/L) | 120.0 (115.0; 124.0) | 126.0 (118.0; 133.0) | 119.0; 124.5 | 0.024 | 0.33 | 503.5 |
| Ferritin (ng/mL) | 20.4 (15.8; 32.9) | 37.0 (18.9; 49.2) | 19.8; 36.6 | 0.055 | 0.28 | 194.5 |
| Zinc (µmol/dL) | 11.1 (9.8; 12.3) | 10.2 (8.2; 11.9) | 9.9; 11.5 | 0.563 | 0.08 | 535.0 |

Data are expressed as median (interquartile range). Between-group differences were assessed using Mann-Whitney U tests; effect size r and BCa 95% CI (2,000 resamples) are reported.

**Table 5. Prevalence of micronutrient deficiencies in children with ASD, by age group ($n = 48$).**

| Indicator | < 5 years ($n = 25$) | 5 – 9 years ($n = 23$) | Total ($n = 48$) | p-value | Risk difference RD (95% CI) |
|---|---|---|---|---|---|
| Anemia | 2/25 (8.0) | 3/23 (13.0) | 5/48 (10.4) | 0.660 | -5.0 (-22.3; 12.3) |
| Iron deficiency | 3/25 (12.0) | 5/23 (21.7) | 8/48 (16.7) | 0.454 | -9.7 (-30.8; 11.4) |
| Low serum zinc | 10/25 (40.0) | 12/23 (52.2) | 22/48 (45.8) | 0.563 | -12.2 (-40.2; 15.8) |
| Anemia + Iron deficiency | 0/25 (0.0) | 2/23 (8.7) | 2/48 (4.2) | NA | -8.7 (-20.2; 2.8) |
| Anemia without iron deficiency | 2/25 (8.0) | 1/23 (4.3) | 3/48 (6.3) | NA | 3.7 (-9.7; 17.1) |
| Iron deficiency without anemia | 3/25 (12.0) | 3/23 (13.0) | 6/48 (12.5) | NA | -1.0 (-19.7; 17.7) |
| One deficiency | 9/25 (36.0) | 10/23 (43.5) | 19/48 (39.6) | 0.474 | -7.5 (-35.1; 20.1) |
| Two deficiencies | 3/25 (12.0) | 5/23 (21.7) | 8/48 (16.7) | 0.432 | -9.7 (-30.8; 11.4) |

NA: Not applicable. p-values were obtained using Fisher's exact test (two-sided). Values are presented as number (%). Risk differences (RD) and 95% confidence intervals were calculated using the Newcombe method (score-Wilson).

**Table 6. Daily nutrient intakes and adequacy among children with ASD (*n* = 48).**

| Nutrient | Intake Median (IQR) | %RDA met Mean ± SD | Intake <75% RDA, n/N (%) |
|---|---|---|---|
| Energy, kJ/d | 4 676 (3 949; 5 199) | 84.6 ± 22.5 | 16/48 (33.3) |
| Protein, g/d | 39.7 (32.5; 49.9) | 159.6 ± 59.2 | 3/48 (6.3) |
| Lipid, g/d | 36.5 (28.5; 51.7) | 113.5 ± 45.4 | 8/48 (16.7) |
| Glucid, g/d | 158.1 (132.8; 174.2) | 82.5 ± 28.4 | 16/48 (33.3) |
| Fiber, g/d | 1.3 (0.9; 1.9) |  | 48/48 (100.0) |
| Calcium, mg/d | 417.5 (288.5; 555.0) | 86.6 ± 85.6 | 27/48 (56.3) |
| Phosphorus, mg/d | 453.0 (327.0; 594.2) | 98.2 ± 46.1 | 17/48 (35.4) |
| Iron (asuming 10%), mg/d | 5.9 (4.6; 8.8) | 111.7 ± 61.1 | 12/48 (25.0) |
| Zinc (moderate bioavailability), mg/d | 3.3 (2.5; 4.0) | 72.7 ± 42.2 | 31/48 (64.6) |
| Iodine, µg/d | 3.6 (2.3; 16.5) | 13.4 ± 28.8 | 47/48 (97.0) |
| Selenium, µg/d | 38.6 (25.5; 50.6) | 188.7 ± 90.2 | 6/48 (12.5) |
| Copper, µg/d | 452.2 (337.5; 542.9) | 102.7 ± 54.5 | 10/48 (20.8) |
| Magnesium, mg/d | 93.7 (63.0; 122.1) | 101.3 ± 62.3 | 18/48 (37.5) |
| Sodium, mg/d | 1118.7 (903.6; 1498.6) | 110.8 ± 52.3 | 23/48 (47.9) |
| Potassium, mg/d | 851.5 (616.4; 1029.2) | 84.9 ± 61.6 | 39/48 (81.3) |
| Vitamin A, µg/d | 267.3 (55.1; 562.8) | 84.1 ± 91.6 | 28/48 (58.3) |
| Vitamin B1, mg/d | 0.66 (0.53; 0.91) | 111.1 ± 68.5 | 10/48 (20.8) |
| Vitamin B2, mg/d | 0.67 (0.46; 0.93) | 100.9 ± 95.3 | 17/48 (35.4) |
| Vitamin PP, mg/d | 6.6 (5.2; 8.4) | 92.6 ± 49.5 | 18/48 (37.5) |
| Vitamin C, mg/d | 21.3 (6.5; 48.2) | 81.7 ± 97.6 | 31/48 (64.6) |

RDA: Recommended dietary allowance. Iron adequacy was calculated assuming 10% bioavailability. Zinc was based on an assumed moderate bioavailability.

## Discussion

This cross-sectional study provides baseline data on the nutritional status of children with ASD in Central Vietnam. Although the study is primarily descriptive, our findings offer empirical evidence that supports a mechanistic pathway: behavioral and physiological traits linked to ASD seem to correspond with a series of impaired dietary habits, which consequently relates to biochemical deficiencies and noted delays in physical growth.

Anthropometric data have shown an increased vulnerability to malnutrition among children with ASD, as evidenced by the coexistence of undernutrition and overnutrition. The prevalence of stunting and underweight in this study was higher than the national estimates reported in Vietnamese children with neurotypical conditions [26]. This increased risk of undernutrition is likely driven by the interplay between socioeconomic constraints and food selectivity. In our cohort, 72.9% of households reported a monthly income below 8 million VND. Low household income may restrict caregivers' ability to provide expensive nutrient-dense foods or specialized dietary alternatives, effectively narrowing the child's food environment to affordable but nutrient-poor staples. Furthermore, the prevalence of overweight and obesity was lower than the national average [26]. While global literature frequently reports a positive correlation between ASD and obesity due to the consumption of ultra-processed foods, our findings indicate a different nutritional trajectory in the Central Vietnam context [27]. Due to the interaction between restrictive eating behaviors and household economic constraints, children with ASD may exhibit a higher propensity for caloric insufficiency rather than caloric excess [28,29]. Consequently, the nutritional risk in this population is mainly shown to be linear growth retardation (stunting) rather than excess weight gain. Thus, nutrition monitoring in children with ASD should adopt a multidisciplinary approach integrating behavioral support with metabolic and nutritional management.

Biochemical assessment revealed that 45.8% of participants were at risk of zinc deficiency, while 16.7% exhibited iron deficiency. These findings correlate with observed dietary patterns, specifically the frequent rejection of animal-source proteins which are primary sources of bioavailable zinc and iron [30]. A critical finding is the high prevalence of constipation and the universal inadequacy of fiber intake. Physiologically, persistent gastrointestinal discomfort might intensify aversions to certain foods, resulting in a limited diet that further reduces the intake of vital nutrients [31]. Zinc plays a crucial role in maintaining the health of the intestinal lining, so a deficiency in this mineral could worsen gastrointestinal issues, establishing a feedback loop where behavioral food avoidance perpetuates physiological problems [32]. These findings support the rationale for a multi-nutrient rather than a single-nutrient intervention strategy. Given the absence of severe deficiencies, a preventive dietary approach is appropriate.

Dietary intake patterns provide a mechanistic context for the observed biochemical deficiencies. While protein intake was largely sufficient, over 50% of the cohort failed to meet 75% of the RDA for calcium, iodine, vitamin C, and zinc. This inadequacy reflects a restrictive dietary phenotype characterized by a reliance on starchy staples and a systematic avoidance of vegetables, fruits, and dairy [33]. Such patterns are consistent with sensory-driven food selectivity documented in ASD populations worldwide, where hypersensitivity to food textures and odors directly limits dietary diversity [3,6]. In the context of Central Vietnam, these dietary gaps are further exacerbated by caregiver and socioeconomic constraints. Given that nearly three-quarters of households reported low income, the intensive effort required to manage ASD-related feeding challenges often leads to a reliance on affordable, but micronutrient-poor foods to ensure basic caloric intake.

Notably, school-aged children exhibited a significantly higher risk of inadequate calcium intake (RD: -33.9%; $p = 0.037$). This age-related decline may be driven by the significant escalation of household economic constraints observed in the older group (87.0% low income), which likely limits the family's capacity to sustain high-cost nutrient-dense foods like dairy. Furthermore, as these children transition to school environments with reduced caregiver oversight, the established sensory-driven food aversions may become more rigid and harder to modify.

There are several strengths to this study. It used a multimodal approach to the assessment, 24-hour recall, food frequency data, and biochemical indicators to capture multiple dimensions of nutritional risk. To address the inherent challenges of a modest sample size, analytical rigor was enhanced through use of bias-corrected and accelerated bootstrap confidence intervals and effect size estimation.

However, several limitations must be acknowledged. First, the small sample size ($n = 48$) constrains the statistical power of the study, increasing the margin of error for prevalence estimates and limiting the precision of our inferences. Due to this logistical constraint, multivariable analysis was not feasible, precluding the adjustment for potential confounding variables such as ASD severity or specific household demographics. Consequently, a definitive causal relationship between ASD symptoms and nutritional deficits cannot be established. Second, the utilization of convenience sampling from specialized intervention centers introduces a notable selection bias. Families accessing these facilities may possess different socioeconomic resources or higher levels of caregiver engagement compared to the broader ASD population in Vietnam. This, combined with the lack of a neurotypical comparison group, limits the generalizability of our findings beyond this specific study setting. Finally, resource constraints prevented the inclusion of other critical biomarkers, such as vitamin D or B12. Despite these limitations, the study provides actionable evidence to guide intervention development and serves as a critical screening phase for the forthcoming trial.

## Conclusion

Children with ASD in Central Vietnam face overlapping nutritional challenges, including stunting and chronic micronutrient inadequacies. These findings provide a context-specific foundation for targeted interventions integrating micronutrient supplementation with behavioral feeding support. While preliminary, this study underscores the necessity of routine nutritional screening to mitigate long-term growth impairments in this vulnerable population.

## Supporting information

**S1 File. Detailed statistical power and precision calculations.** This file details the methodology and formulas used for sample size estimation and power analysis.
(DOCX)

**S2 File. 24-hour dietary recall form.** This form was used to collect detailed information on the types and quantities of foods and beverages consumed by the child during the previous day.
(DOCX)

**S3 File. Food frequency questionnaire.** Semi-quantitative food frequency questionnaire used to assess habitual intake of eight core food groups over the past month.
(DOCX)

**S1 Table. Prevalence of nutrient inadequacy by age group and risk difference (*n* = 48).** Comparison of inadequate intake prevalence across age groups with risk difference and 95% confidence intervals.
(DOCX)

**S2 Table. Frequency of food group consumption among children with ASD (*n* = 48).** Distribution of habitual intake frequencies across eight food groups, categorized by six consumption levels.
(DOCX)

## Author contributions

**Conceptualization:** Huyen Thu Doan, Ngoc Thi Dieu Phan, Ngoc Bao Trinh.

**Data curation:** Huyen Thu Doan, Ngoc Thi Dieu Phan.

**Formal analysis:** Huyen Thu Doan.

**Funding acquisition:** Huyen Thu Doan.

**Investigation:** Huyen Thu Doan.

**Methodology:** Huyen Thu Doan, Ngoc Thi Dieu Phan, Khanh Nam Do.

**Resources:** Ngoc Thi Dieu Phan.

**Software:** Huyen Thu Doan.

**Supervision:** Khanh Nam Do, Oanh Thi Phuong Nguyen, Ngoc Bao Trinh.

**Visualization:** Hoa Thi Ho.

**Writing – original draft:** Huyen Thu Doan, Lieu Thi Thu Nguyen.

**Writing – review & editing:** Huyen Thu Doan, Lieu Thi Thu Nguyen, Khanh Nam Do, Oanh Thi Phuong Nguyen, Ngoc Bao Trinh.

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
