## [Decision Letter · Decision Letter 0]

16 Nov 2025

PGPH-D-25-01847

Micronutrient status and dietary patterns among children with autism in Central Vietnam: A cross-sectional baseline survey to inform targeted intervention

Dear Dr. Huyen,

Thank you for submitting your manuscript to PLOS Global Public Health. After careful consideration, we feel that it has merit but does not fully meet PLOS Global Public Health’s publication criteria as it currently stands. Therefore, we invite you to submit a revised version of the manuscript that addresses the points raised during the review process.

We look forward to receiving your revised manuscript.

Kind regards,

Akim Tafadzwa Lukwa, Ph.D Public Health- Health Economics

Academic Editor

Journal Requirements:

1. Please amend your detailed online Financial Disclosure statement. This is published with the article. It must therefore be completed in full sentences and contain the exact wording you wish to be published.

a) State the initials, alongside each funding source, of each author to receive each grant. For example: “This work was supported by the National Institutes of Health (####### to AM; ###### to CJ) and the National Science Foundation (###### to AM).”

For more information, please go to our submission guidelines:

https://journals.plos.org/globalpublichealth/s/submission-guidelines#loc-financial-disclosure-statement

2. Please ensure that the funders and grant numbers match between the Financial Disclosure field and the Funding Information tab in your submission form. Note that the funders must be provided in the same order in both places as well.

3. Please update your online Competing Interests statement. If you have no competing interests to declare, please state: “The authors have declared that no competing interests exist.”

4. In this instance it seems there may be acceptable restrictions in place that prevent the public sharing of your minimal data. However, in line with our goal of ensuring long-term data availability to all interested researchers, PLOS’ Data Policy states that authors cannot be the sole named individuals responsible for ensuring data access (http://journals.plos.org/globalpublichealth/s/data-availability#loc-acceptable-data-sharing-methods).

5. We do not publish any copyright or trademark symbols that usually accompany proprietary names, eg (R), (C), or TM (e.g. next to drug or reagent names). Please remove all instances of trademark/copyright symbols throughout the text, including ® on page 8.

Additional Editor Comments (if provided):

Dear Authors,

Thank you for submitting this important work. The manuscript addresses a critical evidence gap regarding nutrition among children with ASD in LMIC settings. The combination of anthropometry, dietary recall, FFQ, and biochemical assessments is a major strength, and the paper has clear potential. However, several issues need to be addressed to improve clarity, coherence, and methodological transparency. I outline these below:

Major Comments

Introduction needs restructuring

The current introduction presents useful background but does not clearly articulate:

-The specific research gap,

-The problem statement,

-Why this study is needed in Central Vietnam, and

-How this study builds on and differs from existing Vietnamese literature.

Consider ending the introduction with:

-A concise summary of known gaps,

-A clear statement of objectives or research questions.

Absence of explicit study objectives

Your introduction does not explicitly state what the study set out to measure. Please add specific, numbered objectives such as:

-To assess anthropometric nutritional status…

-To quantify micronutrient biomarkers…

-To describe dietary intake patterns…

-To compare preschool vs. school-aged children…

This will improve alignment between introduction, results, and discussion.

Sampling and study design need clearer description

Reviewer 1’s concern is valid. The manuscript currently does not explain:

-How the five centres were selected (all centres in the province? convenience? purposive selection?).

-Whether the sample is representative of ASD children in the province.

-How many children in total attend these centres, and whether the included 48 reflect 100% enrolment.

-Even with a census approach, transparency is necessary.

Operational definitions missing

Please define:

-Constipation

-Diarrhea

-Inadequate intake (<75% RDA) and explain why this cut-off was chosen

-Low serum zinc thresholds (child age-specific?)

-Overweight/obesity (for 5–9 years, clarify z-score cut-off)

Operational definitions help improve reproducibility.

Analytical rationale requires strengthening

Although statistical methods are technically sound, the rationale for the age-group comparison needs explanation:

-Why under 5 and 5–9 years?

-Was this driven by intervention design? biological differences? programmatic needs?

Also clarify whether any adjustments were considered for:

-Sex

-Socioeconomic status

-Centre location

Even if you ultimately chose descriptive analyses.

Results should be better aligned to stated objectives

-Currently, findings are presented correctly but feel disconnected due to the absence of explicit objectives. After adding objectives, ensure result sections map clearly to them.

Study area description is confusing

Clarify:

-Whether Nghe An province is urban/rural mixed

-Whether intervention centres are government or private

-Whether ASD diagnosis procedures differ across centres

-This contextual information is needed for interpretation.

Minor Comments

-The manuscript is generally readable, but several sentences in the introduction and discussion can be clarified.

-Provide more detail on quality control in laboratory analyses.

-Cite Vietnamese national nutrition statistics for comparison in the results (you cite them in discussion but could bring them earlier).

-Move lengthy statistical calculations (power, precision) to supplementary files to improve flow.

Overall Summary

This is a valuable study with high relevance to global child health and ASD care in LMICs. The dataset is rich and the analysis is sound, but the manuscript requires major revisions to strengthen conceptual framing, methodological clarity, and objective–results linkages. I encourage you to revise the introduction, methods, and structure as outlined above. Addressing these points will significantly improve the scientific contribution and readability of your manuscript.

Reviewers' comments:

Reviewer's Responses to Questions

**Comments to the Author**

1. Does this manuscript meet PLOS Global Public Health’s publication criteria? Is the manuscript technically sound, and do the data support the conclusions? The manuscript must describe methodologically and ethically rigorous research with conclusions that are appropriately drawn based on the data presented.? Is the manuscript technically sound, and do the data support the conclusions? The manuscript must describe methodologically and ethically rigorous research with conclusions that are appropriately drawn based on the data presented.

Reviewer #1: Yes

Reviewer #2: Yes

2. Has the statistical analysis been performed appropriately and rigorously?

Reviewer #1: Yes

Reviewer #2: Yes

3. Have the authors made all data underlying the findings in their manuscript fully available (please refer to the Data Availability Statement at the start of the manuscript PDF file)?

The PLOS Data policy requires authors to make all data underlying the findings described in their manuscript fully available without restriction, with rare exception. The data should be provided as part of the manuscript or its supporting information, or deposited to a public repository. For example, in addition to summary statistics, the data points behind means, medians and variance measures should be available. If there are restrictions on publicly sharing data—e.g. participant privacy or use of data from a third party—those must be specified.requires authors to make all data underlying the findings described in their manuscript fully available without restriction, with rare exception. The data should be provided as part of the manuscript or its supporting information, or deposited to a public repository. For example, in addition to summary statistics, the data points behind means, medians and variance measures should be available. If there are restrictions on publicly sharing data—e.g. participant privacy or use of data from a third party—those must be specified.

Reviewer #1: Yes

Reviewer #2: Yes

4. Is the manuscript presented in an intelligible fashion and written in standard English?

Reviewer #1: Yes

Reviewer #2: Yes

5. Review Comments to the Author

Reviewer #1: I propose the following enhancements to improve the manuscript:

1. The introduction lacks a clear research gap and statement of the problem or scope of study.

2. Specific objectives or research questions are missing in the introduction.

3. Predictor analysis is not adequately addressed.

4. Results should be presented in alignment with specific objectives or research questions.

5. The sufficiency of the sample size for a cross-sectional design is questionable.

6. The manuscript does not detail the sampling method employed.

7. The introduction suffers from a lack of conceptual and theoretical clarity.

8. Operational definitions are absent in the study.

9. For a community-based study, the sampling procedure needs explanation.

10. The description of the study area and sample estimation appears confusing.

Reviewer #2: Good work

Thanks dear authors for your efforts and your work

Well Written manuscript and good idea for research……………………………………………………………

……………………………………………………………………………………………………………………………………………………………………….

6. PLOS authors have the option to publish the peer review history of their article (what does this mean?). If published, this will include your full peer review and any attached files.). If published, this will include your full peer review and any attached files.

**Do you want your identity to be public for this peer review?** For information about this choice, including consent withdrawal, please see our Privacy Policy..

Reviewer #1: No

Reviewer #2: No

Figure Resubmissions:

---

## [Decision Letter · Decision Letter 1]

26 Mar 2026

PGPH-D-25-01847R1

Micronutrient status and dietary patterns among children with autism in Central Vietnam: A cross-sectional baseline survey to inform targeted intervention

Dear Dr. Huyen,

Thank you for submitting your manuscript to PLOS Global Public Health. After careful consideration, we feel that it has merit but does not fully meet PLOS Global Public Health’s publication criteria as it currently stands. Therefore, we invite you to submit a revised version of the manuscript that addresses the points raised during the review process.

As the corresponding author, your ORCID iD is verified in the submission system and will appear in the published article. PLOS supports the use of ORCID, and we encourage all coauthors to register for an ORCID iD and use it as well. Please encourage your coauthors to verify their ORCID iD within the submission system before final acceptance, as unverified ORCID iDs will not appear in the published article. *Only* the individual author can complete the verification step; PLOS staff the individual author can complete the verification step; PLOS staff *cannot* verify ORCID iDs on behalf of authors.verify ORCID iDs on behalf of authors.

We look forward to receiving your revised manuscript.

Kind regards,

Akim Tafadzwa Lukwa, Ph.D Public Health- Health Economics

Academic Editor

Journal Requirements:

Additional Editor Comments (if provided):

Dear Authors,

Thank you for your careful and detailed revision of your manuscript. I appreciate the effort you have taken to engage with the comments provided in the previous round. The manuscript has improved substantially in several important areas, and the revisions are clearly visible.

Strengths of the Revision

I would like to commend you on the following improvements:

-The introduction has been significantly strengthened, with a clearer articulation of the research gap and better contextualization of the study within Vietnam and LMIC settings.

-The addition of explicit, well-structured study objectives greatly improves clarity and alignment across sections.

The methods section is now much more transparent, particularly regarding:

-Sampling approach and centre selection,

-Operational definitions,

Laboratory procedures and quality control.

-The results are now better organised and clearly aligned with the stated objectives.

-Overall, the manuscript is now more coherent, readable, and methodologically transparent.

These revisions have substantially improved the quality of the paper.

Remaining Issues to Address

While the manuscript has improved, there are still a few important issues that need to be addressed before it can be considered for publication.

Strengthen the Conceptual and Analytical Framing

The study remains largely descriptive. While this is understandable given the sample size, the manuscript would benefit from a clearer conceptual structure.

Please provide a brief conceptual framing (either as a short paragraph or simple framework) linking:

-ASD-related feeding challenges

-Dietary intake patterns

-Micronutrient deficiencies

-Anthropometric outcomes

This will help position your findings within a broader scientific understanding rather than presenting them as isolated observations.

Deepen the Discussion and Interpretation

The discussion currently focuses primarily on describing findings rather than interpreting them.

Please strengthen this section by:

-Providing more explanation of why the observed patterns may occur,

-Reflecting on contextual drivers (e.g., feeding behaviour, caregiver constraints, food environment),

-Linking findings more explicitly to existing literature and mechanisms.

Expand the Limitations Section

While some limitations are acknowledged, this section should be strengthened.

In particular, please explicitly discuss:

-The implications of convenience-based centre selection,

-The small sample size (n = 48) and its impact on inference,

-The lack of multivariable analysis, and how this affects interpretation,

-The limited generalisability of findings beyond the study setting.

Minor Clarity Improvements

-Review the manuscript for any remaining minor language inconsistencies.

-Ensure that all sections maintain clarity and conciseness, particularly in the discussion.

Conclusion

This is a valuable and relevant study addressing an important gap in the literature. The dataset is rich, and the revisions have significantly improved the manuscript. With further strengthening of the conceptual framing, interpretation, and limitations, the manuscript will be well-positioned for publication. I encourage you to address the points above in your next revision.

Reviewers' comments:

Reviewer's Responses to Questions

**Comments to the Author**

1. If the authors have adequately addressed your comments raised in a previous round of review and you feel that this manuscript is now acceptable for publication, you may indicate that here to bypass the “Comments to the Author” section, enter your conflict of interest statement in the “Confidential to Editor” section, and submit your "Accept" recommendation.

Reviewer #2: (No Response)

2. Does this manuscript meet PLOS Global Public Health’s publication criteria? Is the manuscript technically sound, and do the data support the conclusions? The manuscript must describe methodologically and ethically rigorous research with conclusions that are appropriately drawn based on the data presented.? Is the manuscript technically sound, and do the data support the conclusions? The manuscript must describe methodologically and ethically rigorous research with conclusions that are appropriately drawn based on the data presented.

Reviewer #2: (No Response)

3. Has the statistical analysis been performed appropriately and rigorously?

Reviewer #2: (No Response)

4. Have the authors made all data underlying the findings in their manuscript fully available (please refer to the Data Availability Statement at the start of the manuscript PDF file)?

The PLOS Data policy requires authors to make all data underlying the findings described in their manuscript fully available without restriction, with rare exception. The data should be provided as part of the manuscript or its supporting information, or deposited to a public repository. For example, in addition to summary statistics, the data points behind means, medians and variance measures should be available. If there are restrictions on publicly sharing data—e.g. participant privacy or use of data from a third party—those must be specified.requires authors to make all data underlying the findings described in their manuscript fully available without restriction, with rare exception. The data should be provided as part of the manuscript or its supporting information, or deposited to a public repository. For example, in addition to summary statistics, the data points behind means, medians and variance measures should be available. If there are restrictions on publicly sharing data—e.g. participant privacy or use of data from a third party—those must be specified.

Reviewer #2: (No Response)

5. Is the manuscript presented in an intelligible fashion and written in standard English?

Reviewer #2: (No Response)

6. Review Comments to the Author

Reviewer #2: Good work

7. PLOS authors have the option to publish the peer review history of their article (what does this mean?). If published, this will include your full peer review and any attached files.). If published, this will include your full peer review and any attached files.

**Do you want your identity to be public for this peer review?** For information about this choice, including consent withdrawal, please see our Privacy Policy..

Reviewer #2: **Yes:** Prof Dr Engy AshaatProf Dr Engy Ashaat

 Figure Resubmissions:

---

## [Editor Report · Decision Letter 2]

14 Apr 2026

Micronutrient status and dietary patterns among children with autism in Central Vietnam: A cross-sectional baseline survey to inform targeted intervention

PGPH-D-25-01847R2

Dear Dr Huyen,

We are pleased to inform you that your manuscript 'Micronutrient status and dietary patterns among children with autism in Central Vietnam: A cross-sectional baseline survey to inform targeted intervention' has been provisionally accepted for publication in PLOS Global Public Health.

Best regards,

Akim Tafadzwa Lukwa, Ph.D Public Health- Health Economics

Academic Editor

Dear Authors,

Thank you for submitting the revised version of your manuscript, “Micronutrient status and dietary patterns among children with autism in Central Vietnam: A cross-sectional baseline survey to inform targeted intervention.” I am pleased to inform you that your manuscript has been accepted for publication in PLOS Global Public Health.

The revisions have substantially strengthened the manuscript in several important ways:

-The introduction now clearly articulates the research gap, problem statement, and study objectives, addressing earlier concerns regarding conceptual clarity and positioning within the existing literature.

-The inclusion of a clear analytical and mechanistic framework linking ASD-related feeding behaviours to nutritional outcomes significantly enhances the scientific contribution and coherence of the paper.

-The methods section has been improved through clearer description of the sampling strategy, study setting, and participant selection, as well as the inclusion of well-defined operational definitions for key variables, improving transparency and reproducibility.

-The results and discussion are now better aligned with the stated objectives, providing a more coherent and interpretable narrative of findings.

-The manuscript makes a valuable contribution by providing integrated anthropometric, dietary, and biochemical evidence on the nutritional status of children with ASD in a low-resource setting an area where data remain scarce.

The study addresses an important and under-explored area at the intersection of nutrition and developmental health in LMICs. Your findings on the coexistence of undernutrition, micronutrient deficiencies, and emerging overnutrition are particularly relevant for informing targeted and context-appropriate interventions.

As a final step, please ensure that all journal formatting and minor editorial requirements are fully addressed prior to production.

Congratulations on this important contribution, and thank you for your thorough and thoughtful revisions throughout the review process.